# Sentence embedding with contrastive multi-views learning

## Abstract

In this work, we propose a self-supervised method to learn sentence representations with an injection of linguistic knowledge. Multiple linguistic frameworks propose diverse sentence structures from which semantic meaning might be expressed out of compositional words operations. We aim to take advantage of this linguist diversity and learn to represent sentences by contrasting these diverse views. Formally, multiple views of the same sentence are mapped to close representations. On the contrary, views from other sentences are mapped further. By contrasting different linguistic views, we aim at building embeddings which better capture semantic and are less sensitive to the sentence outward form.

## 1 Introduction

We propose to learn sentence embeddings by contrasting multiple linguistic representations. The motivation is to benefit from linguistic structures diversity to discard noises inherent to each representation. We aim at encoding high-level representations by aligning the underlying shared information from multiple views.

As illustrated in Figure 1, we train our model with a contrastive framework which aims at mapping close input sentences to close representations while separating unrelated sentences. In Natural Language Processing (NLP), this framework has been widely used to learn word representations (Mikolov et al., 2013a;b) for example. This model relies on the *distributional hypothesis* which conjectures that words within similar context share similar meaning. Such framework has also been extended to sentences with the similar hypothesis that the meaning can be inferred from the context sentences (Logeswaran & Lee, 2018; Kiros et al., 2015).

We propose to extend this framework by assuming that different views of the same sentence should lead to close representation. We considered the dependency trees, a linguistic framework that describes the compositional structure of a sentence. As illustrated in Figure 1, in this framework, the sentence is mathematically described as an oriented acyclic graph where the nodes are words and edges describe the relations between words. Such structure has benefited from an important attention in the NLP community and efficient parser tools for various languages are available, which makes it possible to obtain such information almost freely in the sense it does not require additional *hand annotated* data.

Tree representations are then mapped in a shared embedding space using appropriate Tree LSTM networks introduced in Tai et al. (2015). Model parameters are learned using a discriminating objective as proposed in Logeswaran & Lee (2018).

## 2 Related Work

### 2.1 Representation learning

Representation learning has gained significant attention from the NLP community (Bengio et al., 2013). As illustrated in Figure 3 it is structured in two-steps: (A) a representation is learned using a proxy objective on a usually very large corpora (B) the representation is then used to solve a variety of downstream tasks. Literature comes with a variety of proxy tasks to learn representations. Proxy objectives usually fall in two categories: *(i)* predicting a contextual information or reconstructing

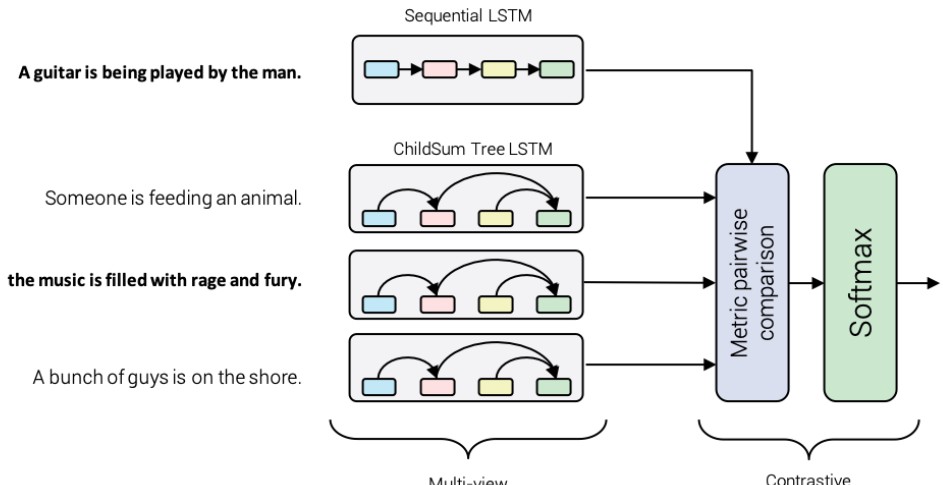

Figure 1: Contrastive multi-views framework. The model is trained to distinguish between different views of context sentences and negative examples. The two views are obtained using a standard LSTM and a Tree LSTM networks on top of a dependency tree structure. A discriminative objective is used to contrast between samples.

an altered input or *(ii)* discriminating multiple representations of the same data among negative examples.

Proxy *(i)* has been declined into multiple variations in the literature. Words embeddings methods Mikolov et al. (2013b) proposes that words with close distributions should lead to close representations. For images, Noroozi & Favaro (2016) proposes to learn representations by solving jigsaw puzzles. With a close taste to NLP tasks, Pathak et al. (2016) aims at learning representations by trying to fill the missing part of an image. van den Oord et al. (2018); Walker et al. (2016) methods aim at predicting the future to represent video or audio .

## 2.2 CONTRASTIVE LEARNING

The second proxy objective *(ii)* is motivated by the observation that end-to-end frameworks tend to learn semantic similarities among classes. For example Wu et al. (2018) reports that the hidden layers of a supervised neural networks assigns close representations to leopard and jaguar images. However an image from lifeboat and leopard will be apart from each other. This motivates for contrastive learning methods where the network aims at distinguishing individual samples to insure a global consistency in the dataset.

Many contrastive learnings have been proved effective in a variety of domains: image processing (Tian et al., 2019; Wu et al., 2018), audio input (van den Oord et al., 2018), sentence representation (Logeswaran & Lee, 2018) or word embedding (Mikolov et al., 2013a;b; Mnih & Kavukcuoglu, 2013).

Besides the nature of data, formalisms might differ on two main aspects: *(i)* How to measure the proximity in both the original and the representation space and *(ii)* What training objective should be used to achieve well distinction between samples.

One possible answer on how to select samples that should lead to close representation is to consider different views of the same sample as proposed in Tian et al. (2019); Li et al. (2018) who propose to combine multi-views of images with contrastive learning. In NLP, Logeswaran & Lee (2018); van den Oord et al. (2018) propose contrastive frameworks for sentences but without multi-views setting.

Multiple metrics to measure the proximity in the representation space are proposed in the literature: Tschannen et al. (2019) enumerates multiple so called *critic* functions. The most straight forward

also used in Logeswaran & Lee (2018) beeing the scalar product: $h_\theta(u, v) = u^T v$. But other critics are considered such as the separable critic $h_\theta(u, v) = \phi_1(u)^T \phi_2(v)$.

The objective function for contrastive learning has also been declined in multiple work: Many relate to Mutual Information (MI) (Hjelm et al., 2019; Tschannen et al., 2019; Bachman et al., 2019) or methods based on triplet loss or max-margin (Chopra et al., 2005; Weinberger et al., 2005).

## 3 METHOD

### 3.1 CONTRASTIVE LEARNING

Contrastive learning is a self-supervised learning method which aims at learning a semantic mapping. Data is embedded with compact representations such that close samples are mapped to nearby points while unrelated ones are affected to apart points. In practice different methods exist to learn such mapping. A straight forward method is to treat each data sample as a distinct class and train a classifier to distinguish between individual instance classes. Such approach is used for word embedding where the goal is to predict a word given his context.

However this is computationally difficult and the outputs are not a finite set in our case since an infinite number of correct sentences might be expressed. A method to approximate a full instance discrimination is to use Negative sampling and Noise Contrastive estimation methods. For every data point $x$ from a dataset $D$ we have access to one paired sample $x^+$ which should be close. We then draw $K$ negative samples $x_1^-, x_2^-, \cdots, x_K^-$ which should be apart. We learn a scoring function $h_\theta$ which assigns large scores to positive paired samples $(x, x^+)$ while low values for negative pairs $(x, x_k^-)$. In practice the function is trained using a classifier which aims at identifying the correct pair $h_\theta(x^+, x^-)$ among negatives pairs $\{h_\theta(x^+, x_k^-)\}_{(k=1,K)}$. In our setup we use a softmax classifier as follow:

$$p(\cdot|x, \{x_1^-, x_2^-, \cdots, x_K^-\}) = \frac{exp(h_\theta(x, \cdot))}{h_\theta(x, \cdot) + \sum_{k=1}^K h_\theta(x_k^-, \cdot)} \tag{1}$$

The classifier is trained using the negative log likelihood loss:

$$L = \sum_{x \in D} logp\left(x^+|x, \{x_1^-, x_2^-, \cdots \in, x_K^-\}\right) \tag{2}$$

As in Logeswaran & Lee (2018) the used scoring function is the inner product $h_\theta(u, v) = u^T v$ in order to avoid the situation where the model learns poor representations but compensates with an excellent classifier on the proxy task. However some authors report excellent results with other *critic* functions: for example Tai et al. (2015) uses a combination of absolute distance and angle measures. Given the scoring vectors for each pair representation we use a simple softmax classifier with a log negative likelihood loss.

### 3.2 MULTI-VIEWS LEARNING

In our setup, the positive pair $(x, x^+)$ is built from different views of the input data. From the original sentences, we construct multiple representations from different views of the object. For every sentence we consider two encoders. $x_s$ which is the representation obtained with a bidirectional LSTM (Hochreiter & Schmidhuber, 1997) who assumes the underlying structure of the sentence to be a sequence, while allowing for long term dependencies. The final representation of the sentence is the concatenation of both direction last state. $x_d$ is obtained with the dependency tree representation combined with a ChildSum Tree LSTM (Tai et al., 2015). The final representation is the state from the root node of the sentence graph. Negative examples are obtained using the dependency Tree LSTM. The dataset $D$ is therefore augmented with the positive examples and multi-views examples leading to $D = \{x_s, x_d, x_{d1}^-, x_{d2}^-, \cdots, x_{dK}^-\}$.

Similarly to Logeswaran & Lee (2018), the model is trained to identify the sentence appearing in the context from the target sentence. The key difference and our contribution is to build different views

of the data. The target sequence is encoded using the sequential Tree LSTM, while the positive and negative samples are encoded using the ChildSum Tree LSTM. As described in the previous section, the objective is to maximize the log probability of the correct sentence to appear in the context.

## 4 EXPERIMENTAL SETUP

### 4.1 DATA

Models are trained on the BookCorpus dataset Zhu et al. (2015). Since the dataset is no longer distributed, a similar dataset is generated using smashword open book data [1]. The obtained dataset contains 78M sentences from 17.000 books. We define a 20k words vocabulary, containing the corpus most frequent tokens. Sentences are tokenized using Microsoft BlingFire tool [2]. Models are trained on a single epoch on the entire corpus without any train-test split. The use of Tree LSTM models supposes to parse sentences in dependency which is operated using the publicly available Spacy parser.

The contrastive learning method supposes to draw negative examples for each sentence pair. We follow the training procedure proposed in Logeswaran & Lee (2018) and build mini-batches respecting books sentences order. Therefore samples from the mini-batch are used as negative examples.

### 4.2 TRAINING PARAMETERS

Models are trained using the Adam optimizer Kingma & Ba (2015) with a 5e-3 learning rate. Gradient is clipped with a value of 5.0. The hidden layer for both Tree LSTM is fixed to 1000 and the embedding dimension to 300. The batch size is 100 and the model is trained on a 1080Ti Nvidia GPU. All model weights are initialized with a xavier distribution Glorot & Bengio (2010) and biases to 0. We observe biases initialization to have a significant impact on the training. Indeed leaf nodes are initialized with zero vectors for state and hidden and initial biases with too significant values tend to slow the training convergence.

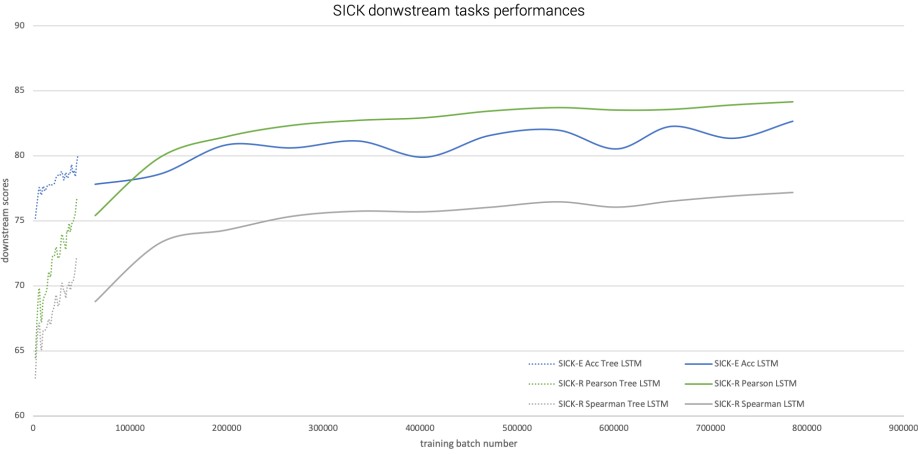

Figure 2: Evolution of performances on the downstream SICK-E (top) and SICK-R (bottom) tasks. A point is computed every hour during the training phase. Despite the batch computation implementation describes in Section A, the Contrastive Tree LSTM is much slower to train. The training was stoped after 33 hours and training on 5.8% of the available training data. The evaluation setup for the SICK tasks is describe in Section 5.

The Tree LSTM network was implemented using a batch procedure described in Section A. However the training was significantly slower than vanilla LSTM. The training phase was stopped after 33

---

[1]Corpus dataset was generated using https://github.com/soskek/bookcorpus
[2]https://github.com/Microsoft/BlingFire

hours of training. The training phase was completed on only 4.6M sentences among the 78M available. We monitored the performance on downstream tasks during training as illustrated in Figure 2.

## 5    RESULTS AND ANALYSIS

As described in Figure 3, the process requires specific evaluation processes at each learning step. Although methods are developed to control the properties of the intermediate representation such as probing tasks for NLP Conneau et al. (2018), the intermediate representation is usually evaluated on his ability to solve the final task: downstream evaluation. To facilitate the comparison across multiple representations and limit the impact of the downstream algorithm, the linear evaluation protocol (Kiros et al., 2015) is usually used. It proposes to solve the downstream task with a minimal and simple model: a logistic regression. At test time, the concatenation of two encoders is used.

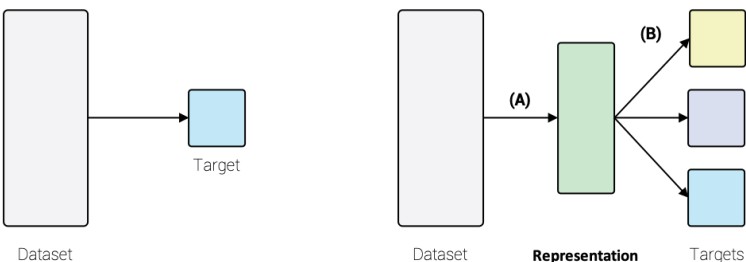

Figure 3: (left) End-to-end framework where the algorithm learns to map directly the inputs to the target. The hidden sates of the neural network are only used for this task. (right) In contrast to end-to-end approaches the algorithm proceeds in two phases. First a representation is learned with a self-supervised proxy task. The same representation might then be used for different tasks.

### 5.1    DOWNSTREAM EVALUATION

Sentence representations are evaluated on downstream tasks that require to capture the underlying sentence semantic. 7 classification tasks from the SentEval benchmark are applied: movie review sentiment (**MR**) (Pang & Lee, 2005), product reviews (**CR**) (Hu & Liu, 2004), subjectivity classification (**SUBJ**) (Pang & Lee, 2004), opinion polarity (**MPQA**) (Wiebe et al., 2005), question type classification (**TREC**) (Voorhees, 2003) , semantic relatedness and entailment (**SICK-R, SICK-E**) (Marelli et al., 2014) and paraphrase identification (**MRPC**) (Dolan et al., 2004).

The MR, CR, SUBJ, MPQA tasks are binary classification tasks with no pre-defined train-test split. A 10-fold cross validation is used in reporting test performance for these tasks. For other tasks we use the proposed train/dev/test splits. The dev set is used for choosing the regularization parameter and results are reported on the test set. We follow the linear evaluation protocol of Kiros et al. (2015) where a logistic regression classifier is trained on top of sentence representations with the cross-validation procedure described earlier. This protocol facilitates the comparison across studies and avoids the case of a good classifier which compensates for bad sentence representations. The script from Conneau & Kiela (2018) is used for downstream evaluation.

The scores are not as good as the state of the art results reported in Table 1. However the model was not trained on the entire available training set used in Logeswaran & Lee (2018) . The model shows encouraging properties and trends in Figure 2 suggest it might improve with a larger exposition to the training data.

| Model | MR | CR | SUBJ | MPQA | TREC | SICKR | SICKE | MRPC |
|---|---|---|---|---|---|---|---|---|
| **Supervised training methods** | | | | | | | | |
| GloVe Bow | 78.1 | 80.4 | 91.9 | 87.8 | 85.2 | 76.4 | — | **81.1** |
| LSTM | — | — | — | — | — | 85.3 | — | — |
| BidirLSTM | — | — | — | — | — | 85.7 | — | — |
| Tree-LSTM | — | — | — | — | — | 86.8 | — | — |
| Infersent | **81.1** | **86.3** | **92.4** | **90.2** | **88.2** | **88.4** | **86.1** | 76.2 |
| **Self-supervised training methods** | | | | | | | | |
| Fastsent | 71.8 | 78.4 | 88.7 | 81.5 | 76.8 | — | — | 80.3 |
| Skip-Thought | 76.5 | 80.1 | 93.6 | 87.1 | 92.2 | 85.8 | — | 82.0 |
| Dissent | 80.2 | 85.4 | 93.2 | **90.2** | 91.2 | 84.5 | 83.5 | 76.1 |
| Quick-thoughts | **82.4** | **86.0** | **94.8** | **90.2** | **92.4** | 87.4 | — | **84.0** |
| Contrastive Tree | 64.0 | 72.8 | 86.8 | 77.7 | 82.4 | 76.8 | 80.0 | 79.1 |

Table 1: Comparison of sentence representations on downstream tasks. For the SICK-R task, the pearson coefficient is indicated. For the MRPC task, this is the F1 score. The baseline is reported from (Logeswaran & Lee, 2018) and is obtained using a bag-of-words representation. LSTM, BidirLSTM and Tree LSTM are reported from Tai et al. (2015). Fastsent is reported from (Logeswaran & Lee, 2018), Dissent and Infersent from (Nie et al., 2019), Skip-thoughts from (Kiros et al., 2015) and Quick-thoughts from (Logeswaran & Lee, 2018). The table is divided into different sections. The bold-face numbers indicate the best performance values among models in the current and all previous sections. Best overall values in each column are underlined.

## 5.2 PROBING TASKS

Conneau et al. (2018) argues that downstream tasks require complex form of inference, which makes it difficult to assess the fine grained quality of a representation. *Probing tasks* are supposed to overcome this limitation and separately evaluate individual linguistic properties of representations. The probing benchmark includes surface information tasks: predicting the length of the sentence (**SentLen**) and the word content (**WC**). Syntactic tasks include predicting the depth of the sentence tree (**TreeDepth**), the root top constituents (**TopConst**) and the word order (**BShift**). Finally semantic tasks aim at predicting the verb tense (**Tense**), if the subject and complement are singular or plural (**SubjNum, ObjNum**) and semantic or coordination inversion (**SOMO, CoordInv**). The probing results are presented in Table 2 for both Contrastive LSTM (CL) and Contrastive Tree LSTM (CTL).

| Probing task | LSTM (CL) | Contrastive Tree LSTM (CTL) |
|---|---|---|
| **SentLen** | 74.7 | **83.0** |
| **WC** | **66.4** | 23.1 |
| **TreeDepth** | **35.1** | 33.4 |
| **TopConst** | **81.2** | 77.3 |
| **Bshift** | **59.7** | 58.1 |
| **Tense** | **87.7** | 82.5 |
| **SubjNum** | **79.8** | 74.5 |
| **ObjNum** | **74.4** | 66.0 |
| **SOMO** | **52.3** | 50.5 |
| **CoordinV** | **67.9** | 63.4 |

Table 2: Probing task scores for the Contrastive Tree LSTM (CTL) and the Contrastive LSTM (CL). Bold-face numbers indicate the best performance values among the two models. The proposed train-test split is used for each task. Parameters are fixed on the dev set and results are presented on the test set. Results for the Contrastive LSTM (CL) are obtained using a model we trained on our version of the bookcorpus as described in Section 4.

We observe the interaction between the standard Tree LSTM and the Tree LSTM is sensible to the sentence length which was also observed by (Tai et al., 2015). All other scores are bellow the results

obtained by the application of the sole LSTM, pointing that the introduction of linguistic knowledge does not compensate enough for the smaller training set.

### 5.3 QUALITATIVE ANALYSIS

Our training setup and Logeswaran & Lee (2018) share a similar framework but with different models to embed sentences. We retrieved the nearest neighbor from several query examples with the two methodologies. The sentences are extracted from the SICK test set. The closest neighbors are determined using the cosine similarity and presented in Table 3. Examples are selected to illustrate the ability of models to capture desired linguistic properties such as passive form, concepts, counting faculties or gender identification.

| Top 5 sentences LSTM | Score | Top 5 sentences Tree LSTM | Score |
|---|---|---|---|
| **The guitar is being played by the man** | | | |
| A guitar is being played by the man | 0.046 | A guitar is being played by the man | 0.036 |
| The man is playing an acoustic guitar | 0.054 | A guitar is being played by a man | 0.064 |
| The man is playing the guitar | 0.055 | A guitar is being passionately played by a man | 0.071 |
| The man is a guitar player | 0.057 | The guitar player loves the man passionately | 0.120 |
| The guitar player loves the man passionately | 0.059 | The guitar is being played by the man , who has the guitar case open for donations | 0.123 |
| **A woman is dancing on a stage** | | | |
| A woman is dancing | 0.019 | A young girl is dancing | 0.083 |
| A woman is dancing gracefully | 0.024 | A woman is washing a big pepper | 0.092 |
| A woman is dancing beautifully in a cage | 0.050 | The woman is not seasoning the oil | 0.096 |
| A woman is dancing and a man is playing the keyboard | 0.058 | A little girl is playing a piano , which is very big , on stage | 0.099 |
| A woman is staging a dance | 0.073 | A woman is dancing | 0.102 |
| **A man and woman are talking** | | | |
| A man and a woman are speaking | 0.048 | A man and a woman are speaking | 0.014 |
| A man and a woman are sitting | 0.077 | A man is talking to a woman | 0.052 |
| A man is talking to a woman | 0.093 | The man and woman are not walking | 0.082 |
| A woman is ignoring a man | 0.155 | One man is talking to a girl with an internet camera | 0.083 |
| A man is talking | 0.156 | A man and a woman are walking | 0.086 |
| **A bunch of guys is on the shore** | | | |
| A group of people is near the ocean | 0.134 | A bunch of guys is on the shore | 0.012 |
| Two men are standing in the ocean | 0.189 | There is no one on the shore | 0.081 |
| Four people are floating on a raft | 0.202 | A person is rinsing a steak with water | 0.092 |
| A crowd of people is near the water | 0.213 | A sea turtle is not hunting for fish | 0.098 |
| Two men are walking through a river | 0.238 | A sea turtle is hunting for fish | 0.103 |

Table 3: Nearest neighbors retrieved by the Contrastive Tree LSTM (CTL) and the Contrastive LSTM (CL). For each sentence, the 5 closest neighbors from the test split of the SICK dataset are retrieved using the cosine similarity from their representations. Results are presented in decreasing order of cosine distance.

Both models might be fooled with surface information and retrieve examples with similar words but different meanings. Sole LSTM encoder presents remarkable properties to identify passive form. The Tree LSTM presents interesting properties such as retrieving sentences with similar principal

proposition but complement which do not alter the sense of the sentence. Tree LSTM also captures number properties and identifies concepts such as 4 people might be referred as a group.

## 6 CONCLUSION

We exploit the diversity of linguistic structures to build sentence representations. Out method shows promising results and does not require hand annotated data. More scalable implementations might be considered to explore more experimentation setups. Although results are below state of the art performances, our model is trained on only a small proportion of the bookcorpus sentences as stated in Figure 2. A larger exposition to the training data and an extended training time might benefit to the downstream and probing scores. Other linguistic structures might also be tested such as constituency tree associated with N-ary Tree LSTM or Tree LSTM improved with an attention mechanism.

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

# A   COMPUTING METHOD FOR TREE LSTM

We implemented a batching procedure to fasten Tree LSTM computations. Group of nodes are computed sequentially to insure all node children have already been computed. Nodes are considered given their distance to the root node. First, Leaf nodes with highest depth are computed to progressively compute inner nodes. The Tree LSTM cell implementation is specifically designed to treat simultaneously all nodes in each the step.

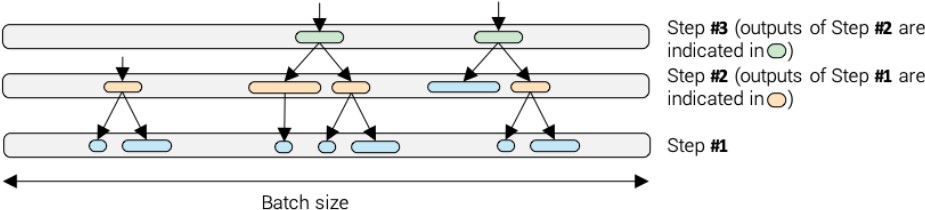

Figure 4: The batching procedure to optimize the graph computation. For each batch, the computation is decomposed in steps which insure that every node dependent have already been computed. At each step, node with the same depth to the root are computed in a single operation and the output fed to the next computational step.

