# OpenReview forum: "Sentence embedding with contrastive multi-views learning"
_ICLR.cc/2020/Conference — Reject_

### Official Review · AnonReviewer2 · 2019-10-16
**Official Blind Review #2**

**Rating:** 1

**Review:**

The paper proposes a new sentence embedding method. The novelty is to use dependency trees as examples in the self-supervised method based on contrastive learning. The idea to use linguistic knowledge in the design of sentence embeddings is attractive. The sentence representation is computed by a bi-LSTM and dependency tree representations are computed by Tree LSTM. The softmax classifier is trained using the negative log-likelihood loss.

In my opinion, the paper could not be accepted. As said before, the idea is attractive but the paper lacks motivations for the choice of dependency trees as additional linguistic knowledge. Indeed, the goal of the proposed algorithm should be made more precise. It is, in my opinion, very difficult to do better than existing sentence embedding methods and the proposed method should be used for specific downstream tasks where the structure of sentences is meaningful. Moreover, the proposed method do not scale well and empirical results on classical downstream tasks are not convincing. Last, in my opinion, the redaction of the paper should be improved and the bibliography should be updated. For instance, the best up-to-date sentence embedding methods are not cited (ELMO and BERT).

Detailed comments.

* Abstract and introduction. The description of the contribution is not precise enough. Please make precise what are "multiple views", "different linguistic views". Please explain why you choose dependency trees and explain why their use can improve sentence embeddings.
* Related work. Please consider only word embeddings and sentence embeddings because the literature is sufficiently large in the last few years. Please update your related work with methods such as ELMO and BERT and subsequent work. Also recent papers study how BERT embeddings embed structural information and these should be discussed as you consider dependency trees in the construction of sentence embeddings.
* The method does not scale well. The paper does not propose ideas to solve this problem. Why don't you consider the approach used in Logeswaran et al.
* The qualitative analysis shows that similar sentences have a similar structure. This is not surprising because dependency trees are used for learning. But this should give ideas of downstream tasks for which the approach could be fruitful.
* Many typos.

**Experience Assessment:**

I have published one or two papers in this area.

**Review Assessment: Checking Correctness Of Derivations And Theory:**

I assessed the sensibility of the derivations and theory.

**Review Assessment: Checking Correctness Of Experiments:**

I did not assess the experiments.

**Review Assessment: Thoroughness In Paper Reading:**

I read the paper at least twice and used my best judgement in assessing the paper.

---

### Official Review · AnonReviewer1 · 2019-10-23
**Official Blind Review #1**

**Rating:** 1

**Review:**

Overview: This work proposes to learn sentence embeddings using both contrastive learning and multiple "views" of sentences.  This work largely builds off of [1], including using the same objective, but uses a multi-view approach to modeling.
    - They apply the concept of multi-view models, specifically combining tree and linear LSTMs to learning sentence representations.
    - They prepare a new, large-scale book dataset, which is useful because the previously commonly used book dataset was taken down for legal reason.
    - They provide a fairly broad set of analyses on their model, both quantitative and qualitative, performance-driven and analysis-driven.
- Review: The ideas and models presented in this paper are not new, while the supporting experiments are not very well done or convincing. Overall, I recommend rejecting this work.
- The models are contrastively learned in that they are trained to embed "similar" sentences nearby in the embedding space, and "dissimilar" sentence far away, where "similar" sentences are defined as consecutive sentences. This method of learning textual representations is well-established in the NLP literature, mostly prominently in recent years with word embedding models like Skip-Gram and in sentence embedding models like in [1], [2], [3] (the next sentence prediction task), and several more.
- In practice, the multiple views of each sentence that this paper considers boils down to encoding the sentence with a bidirectional LSTM and a TreeLSTM and concatenating the representations from each encoder. This idea again has been established in the literature ([4], [5], [6]).
- The experiments don't seem setup to demonstrate that the multiple views are beneficial over a single view. In Table 1, there are rows for just an LSTM or just a TreeLSTM, but they seem to be trained with labeled data whereas the proposed method is trained self-supervised. A more informative comparison to demonstrate the value of using multiple views would be to train the LSTM and TreeLSTM with the same objective (and ideally model size). Overall, I don't think the claims in the paper are well-supported by the model proposed or the experiments.
- I have a number of concerns about the experiments.
    - "Models are trained on a single epoch on the entire corpus without any train-test split": so there is no early stopping? Why stop training after one epoch? Was there any indication you were overfitting the data?
    - "The training phase was stopped after 33 hours of training": Why stop there? Computational constraints? Later comments suggest this is quite premature ("training phase was completed on only 4.6M sentences among the 78M available").
    - The results seem to indicate that this method underperforming recent work significantly.

Areas of improvement
- Some of the language in the introduction and conclusion are a bit of a stretch. Using a linear and tree LSTM (based on dependency parses) doesn't really represent a "diversity of linguistic structures".
- Related work: There's no mention of pretained language models, which could be seen as a form of representation learning for language, and have been hugely impactful in NLP.
- Method
    - Missing negative in the log likelihood
    - Why do you use inner product if other works "report excellent results" with other scoring functions?
    - "assumes the underlying structure of the sentence to be a sequence, while allowing for long term dependencies": If anything, the treeLSTM more easily allows for long-term dependencies than the linear LSTM.
    - "Negative examples are obtained using the dependency Tree LSTM": I'm not totally sure how the negatives are obtained here.
    - "The target sequence is encoded using the sequential Tree LSTM, while the positive and negative samples are encoded using the ChildSum Tree LSTM": why are the sentences not all encoded with the same encoder?
    - It looks really odd that most of Table 1 is empty. Given your model, I imagine it can't have been that difficult to evaluate more baselines (BiLSTM and TreeLSTM) on the rest of the tasks.
    - It'd be nice if you could clearly indicate in Table 1 which method is yours.
- Results and Analysis
    - The standard evaluation setting for sentence embeddings would be GLUE or SuperGLUE.
    - A glaringly missing baseline is BERT (or any of its relatives), which is also self-supervised.
    - The results are underwhelming, and as the author admits, somewhat premature as training didn't seem to finish.
    - 5.2: what are the contrastive LSTM and Tree LSTM? Are those the learned encoders from the "Contrastive Tree" in Table 1, or are they trained from scratch?
    - I don't think the analyses in Sections 5.2 and 5.3 or Figure 2 are particularly useful.
- There are a noticeable number of typos. For example, in the abstract: "this linguist[ic] diversity" and "better capture semantic[s]". It'd be worthwhile to look over the paper closely for typos.


[1] AN EFFICIENT FRAMEWORK FOR LEARNING SENTENCE REPRESENTATIONS. Lajanugen Logeswaran and Honglak Lee
[2] Discourse-Based Objectives for Fast Unsupervised Sentence Representation Learning. Yacine Jernite, Samuel R. Bowman, David Sontag
[3] BERT: Pre-training of Deep Bidirectional Transformers for Language Understanding. Jacob Devlin, Ming-Wei Chang, Kenton Lee, Kristina Toutanova
[4] Enhancing and Combining Sequential and Tree LSTM for Natural Language Inference. Qian Chen, Xiaodan Zhu, Zhenhua Ling, Si Wei, Hui Jiang.
[5] Enhanced LSTM for Natural Language Inference. Qian Chen, Xiaodan Zhu, Zhenhua Ling, Si Wei, Hui Jiang, Diana Inkpen.
[6] Improving Sentence Representations with Consensus Maximisation. Shuai Tang, Virginia R. de Sa.

**Experience Assessment:**

I have published in this field for several years.

**Review Assessment: Checking Correctness Of Derivations And Theory:**

I assessed the sensibility of the derivations and theory.

**Review Assessment: Checking Correctness Of Experiments:**

I assessed the sensibility of the experiments.

**Review Assessment: Thoroughness In Paper Reading:**

I read the paper at least twice and used my best judgement in assessing the paper.

---

### Official Review · AnonReviewer4 · 2019-11-03
**Official Blind Review #4**

**Rating:** 3

**Review:**

This paper describes a self-supervised sentence embedding approach that incorporates a different view from plain text where some extent of linguistic knowledge is incorporated through the application of tree LSTM. The training procedure is standard contrastive framework where the model is encouraged to distinguish between context sentence (sentences appearing close to the target sentence) and negative samples. Evaluations are conducted on 1) downstream tasks, but with a simple logistic regression model on top of sentence embeddings; 2) probing tasks that more focus on surface information prediction, syntactic and semantic tasks; and 3) qualitative analysis with nearest 5 sentences.

Although the experiments are thorough, I am in favor of rejecting this paper with the following reasons:

First, the proposed model is trained with 4.6M sentences among 78M available for 33 hours. It is unclear why authors stop the training at this early stage but the results on all three evaluations seem to be inferior to the state-of-the-art by a big margin. I am happy to raise my score if authors can show the results of a well trained proposed model.

Second, the paper has some room for improvement in terms of clarity, to name a few:
1) Authors can strengthen the motivation for multi-views learning in related work;
2) Formula 1 for softmax is wrong;
3) Contrastive LSTM and contrastive tree LSTM are not clearly defined in the paper, although the former should refer to quick-thoughts and the latter means the proposed method;
4) In qualitative analysis, for the last example, there is exactly the same candidate with similarity score 0.012. According to cosine similarity, wouldn’t this be 0 and also show up in the baseline model regardless of the embeddings?

**Experience Assessment:**

I have published one or two papers in this area.

**Review Assessment: Checking Correctness Of Derivations And Theory:**

I carefully checked the derivations and theory.

**Review Assessment: Checking Correctness Of Experiments:**

I carefully checked the experiments.

**Review Assessment: Thoroughness In Paper Reading:**

I read the paper thoroughly.

---

### Decision · Program_Chairs · 2019-12-19

**Decision:**

Reject

**Comment:**

This paper proposes a method to learn sentence representations that incorporates linguistic knowledge in the form of dependency trees using contrastive learning. Experiments on SentEval and probing tasks show that the proposed method underperform baseline methods.

All reviewers agree that the results are not strong enough to support the claim of the paper and have some concerns about the scalability of the implementation. They also agree that the writing of the paper can be improved (details included in their reviews below).

The authors acknowledged these concerns and mentioned that they will use them to improve the paper for future work, so I recommend rejecting this paper for ICLR.